# Targeting Viral Transcription for HIV Cure Strategies

**DOI:** 10.3390/microorganisms12040752

**Published:** 2024-04-08

**Authors:** Jon Izquierdo-Pujol, Maria C. Puertas, Javier Martinez-Picado, Sara Morón-López

**Affiliations:** 1IrsiCaixa, 08916 Badalona, Spain; jizquierdo@irsicaixa.es (J.I.-P.); mcpuertas@irsicaixa.es (M.C.P.); jmpicado@irsicaixa.es (J.M.-P.); 2Germans Trias i Pujol Research Institute (IGTP), 08916 Badalona, Spain; 3CIBERINFEC, 28029 Madrid, Spain; 4Department of Infectious Diseases and Immunity, School of Medicine, University of Vic-Central University of Catalonia (UVic-UCC), 08500 Vic, Spain; 5Catalan Institution for Research and Advanced Studies (ICREA), 08010 Barcelona, Spain

**Keywords:** viral persistence, HIV, HIV transcription, HIV nuclear export, HIV latency, latency-reversing agents, latency-promoting agents

## Abstract

Combination antiretroviral therapy (ART) suppresses viral replication to undetectable levels, reduces mortality and morbidity, and improves the quality of life of people living with HIV (PWH). However, ART cannot cure HIV infection because it is unable to eliminate latently infected cells. HIV latency may be regulated by different HIV transcription mechanisms, such as blocks to initiation, elongation, and post-transcriptional processes. Several latency-reversing (LRA) and -promoting agents (LPA) have been investigated in clinical trials aiming to eliminate or reduce the HIV reservoir. However, none of these trials has shown a conclusive impact on the HIV reservoir. Here, we review the cellular and viral factors that regulate HIV-1 transcription, the potential pharmacological targets and genetic and epigenetic editing techniques that have been or might be evaluated to disrupt HIV-1 latency, the role of miRNA in post-transcriptional regulation of HIV-1, and the differences between the mechanisms regulating HIV-1 and HIV-2 expression.

## 1. Introduction

An estimated 39 million people worldwide are living with HIV, 1.7 million of them being children from 0 to 14 years old and 2 million of them living with HIV-2. With an incidence of 1.3 million new infections in 2022, the HIV pandemic remains an important global health problem [1]. Since the introduction of antiretroviral therapy (ART), the infection has transitioned from a lethal to a chronic disease, and the incidence has been gradually decreasing [2]. Although ART controls HIV replication and dramatically reduces the chance of developing AIDS, the availability of a universal cure for HIV seems a little bit far. This is mainly due to the rapid establishment of viral reservoirs that escape the immune response and, therefore, remain latent. Due to these viral reservoirs, most ART-suppressed people living with HIV (PWH) after ART discontinuation would experience viral rebound, forcing them to be on ART indefinitely.

Understanding the mechanisms that control HIV expression will provide insights into how HIV latency, pathogenesis, and persistence are regulated. In particular, HIV latency, which is a major barrier to a cure, is maintained by combinatorial mechanisms at transcriptional and post-transcriptional levels.

The disruption of viral transcription is postulated as a potential target for therapeutic intervention, among the different strategies aimed at curing HIV [3]. Specifically, by activating the expression of latent proviruses, known as “shock and kill”, or by promoting the silencing of HIV transcription, known as “block and lock”. On the one hand, pharmacological induction of latent HIV transcription began with the use of epigenetic modulators, with the hope of generating viral proteins that could activate specific cytotoxic responses to reduce the viral reservoir [4,5]. Subsequently, many other pharmacological compounds with different mechanisms of action have been proposed using the same strategy as pharmacological latency-reversing agents (LRAs) [6]. However, they mainly differ in the stage of HIV transcription that they target to reactivate the latently infected cell. On the other hand, the mechanistically opposite strategy, which pursues the definitive silencing of HIV transcription, includes a more limited number of latency-promoting agents (LPAs) with a lower diversity in their mechanisms of action [7,8,9,10,11,12,13,14,15].

In conclusion, HIV cure strategies based on the disruption of viral transcription are in constant evolution and represent the scientific focus of multiple experimental interventions addressed to impact HIV latency. Here, we review the cellular and viral factors that regulate HIV transcription and post-transcription mechanisms, their pharmacological interference using novel approaches to tackle viral latency, and their parallels with HIV-2.

## 2. Cellular and Viral Factors Regulating HIV Transcription and Nuclear Export

A summary of the cellular and viral factors regulating HIV transcription stages (initiation, elongation, polyadenylation, and multiple splicing) and nuclear export are described in the following sections and represented in Figure 1.

### 2.1. Initiation

HIV-1 transcription initiation starts in the trans-activator response element (TAR) of the 5′ long terminal repeat region (LTR) and is dependent on different cellular host factors. Transcription initiation is mainly controlled by cis-regulatory elements along 5′LTR. These cis-regulatory elements are recognized by either constitutively expressed transcription factors such as specificity protein 1 (Sp1) [16,17] and o-binding protein 1 (Oct1) [18], and inducible transcription factors such as nuclear factor-kappa B (NF-κB) [19,20], activator protein 1 (AP-1) [16,17], and the nuclear factor of activated T cells (NFAT) [16,17]. Therefore, the initiation of viral transcription depends on the balance between transcription factors that are activators or repressors and their binding to the 5′LTR loop. This way, only when activator transcription factors are bound to the viral cis-regulatory elements will the cellular RNA polymerase II (RNAPII) machinery be recruited and halted until the elongation step begins [21].

### 2.2. Elongation

The elongation process is mainly driven by human positive transcriptional elongation factor b (P-TEFb) and the viral transactivator Tat. Tat and P-TEFb, through many different steps (reviewed in [21]), are responsible for the phosphorylation of the carboxy terminal domain (CTD) of RNAPII, thereby switching the pausing state of RNAPII to an active elongation [22]. Moreover, P-TEFb and Tat undergo multiple cycles of association/dissociation, allowing potent expression of the HIV-1 genome by repeating their function in a loop [23].

### 2.3. Polyadenylation

HIV mRNA follows the same polyadenylation process as other cellular mRNAs. The polyadenylation process starts with an endonucleolytic cleavage by CPSF3. Then, the poly(A) polymerase (PAP) finishes the process by adding 100 to 250 adenylate residues (reviewed in [24]). In order for this process to start, CPSF3 needs to recognize the cleavage site of the mRNA. This cleavage site consists of an AAUAAA or similar hexanucleotide and a 25–30 base short U- or GU-rich zone that can be located either downstream or upstream of the hexanucleotide. Among the elements that participate in this process, there are 13 core elements and almost 80 auxiliary elements that, together with the core elements, are responsible for the polyadenylation of mRNAs [25]. Moreover, HIV has two polyadenylation sites, one at each LTR region, either 5′ or 3′ [26]. It is essential that the virus suppresses the upstream 5′ polyadenylation signal, or there will be no open reading frame for some mRNAs, constituting one of the mechanisms promoting HIV latency [27] and, therefore, an interesting therapeutic target (reviewed in [24]).

### 2.4. Multiple Splicing

Splicing in HIV is fundamental in order to produce competent viral particles. HIV-1 first needs to generate a 9 kb unspliced vRNA (HIV-1 US vRNA) encoding the full viral genome, which will serve as the genomic RNA that will be packaged in new virions. But still, splicing in HIV-1 is essential in order to produce both structural components of the viral particles and accessory proteins. Thus, this 9 kb long transcript is processed by the spliceosome cellular machinery and subjected to alternative splicing that will produce either the 4 kb single spliced vRNA (HIV-1 SS vRNA) or the 2 kb multiply spliced vRNA (HIV-1 MS vRNA) (reviewed in [28]). HIV-1 MS vRNAs are basically produced in the early stages of HIV transcription since they encode for Tat, which enhances HIV transcription initiation, and Rev, which will be responsible in part for the translocation of HIV-1 US vRNA from the nucleus to the cytoplasm [28]. It has been recently reported that PCID2 establishes blocks to alternative splicing during HIV-1 latency and misregulates alternative splicing in cells obtained from PWH [29]. However, there is limited knowledge of the specific cellular molecules and proteins involved in HIV splicing. In trying to better understand the host factors involved in HIV-1-splicing, a recent study showed several genes upregulated and downregulated in association with HIV splicing [30]. Among them, four were involved specifically in the minor mRNA splicing pathway [31,32,33], of which three were upregulated (RNAU4ATAC, SNRNP25, SNRPD2) and one was downregulated (RNAU4ATAC11P). These results suggested that the minor mRNA splicing pathway might be involved in HIV splicing. Therefore, disruption of the HIV-1 splicing process is also an interesting therapeutic approach for both LRAs and LPAs.

### 2.5. Nuclear Export

Each of the three types of vRNA resulting from splicing (HIV-1 US, SS, or MS vRNA) uses a specific mechanism for its translocation from the nucleus to the cytoplasm. During the early phases of HIV transcription, the shortest HIV-1 MS vRNA transcripts are is exported through the nuclear pores via the canonical NXF1/NXT1-mediated export [34,35]. This MS vRNA will produce Rev, a protein that will be key for the export of both SS and US HIV-1 vRNA. Rev contains a nuclear localization signal (NLS) that allows it to return to the nucleus by interacting with Importin B [36]. It will then be bound to the Rev response element (RRE) and mediate the long vRNA export via a CRM-1-dependent mechanism [37,38]. Therefore, halting the export of MS vRNA in order to abrogate Rev function or directly block the export of SS and US vRNA from the nucleus to the cytoplasm are potential therapeutic approaches.

### 2.6. Repressive Transcriptional Factors

There are multiple host transcriptional repressor factors known to be involved in the negative regulation of HIV transcription [21,39]. In this context, several studies have demonstrated that the restriction factor TRIM22 impairs HIV-1 transcription [40,41,42,43,44]. Specifically, it inhibits the binding of Sp1 to the HIV-1 promoter by forming protein complexes with other cellular proteins that bind and sequester Sp1 [45]. Another example is the COUP-TF interacting protein 2 (CTIP2), also known as BCL11B, which represses HIV-1 transcription initiation by establishing a repressive epigenetic environment in the LTR [46] and by repressing TAT-mediated transactivation in a complex with NuRD [47]. Another study showed that KLF2 and KLF3 repressed HIV-1 and HIV-2 transcription by direct binding to the LTR [48]. Moreover, the negative elongation factor (NELF) induces RNAP II promoter-proximal pausing, limiting HIV expression [49]. It has also been shown that ZBTB2 binds the HIV-1 promoter, recruiting histone deacetylases (HDACs) and repressing HIV-1 transcription [50]. Furthermore, it has been recently reported that PCI domain-containing 2 (PCID2) protein, a subunit of the transcription and export complex 2 (TREX2) complex, binds to the latent HIV-1 LTR as a transcriptional repressor [29]. Moreover, PCF11, a protein involved in 3′ end processing of mRNA and transcription termination of protein-encoding genes [51,52], causes premature termination of HIV-1 transcription [53]. Also, a study recently showed that WDR82 associates with PCF11 at a proximal RNAPII elongation checkpoint on the HIV-1 promoter, enforcing premature transcription termination [54]. Furthermore, another recent study observed that the epigenetic repressor TASOR, a protein of the Human Silencing Hub (HUSH) complex, cooperates with the RNA deadenylase CCR4–NOT complex scaffold CNOT1 and synergistically represses HIV-1 expression [55]. Therefore, increasing the expression of these transcriptional repressor factors may also promote HIV-1 latency.

**Figure 1 microorganisms-12-00752-f001:**
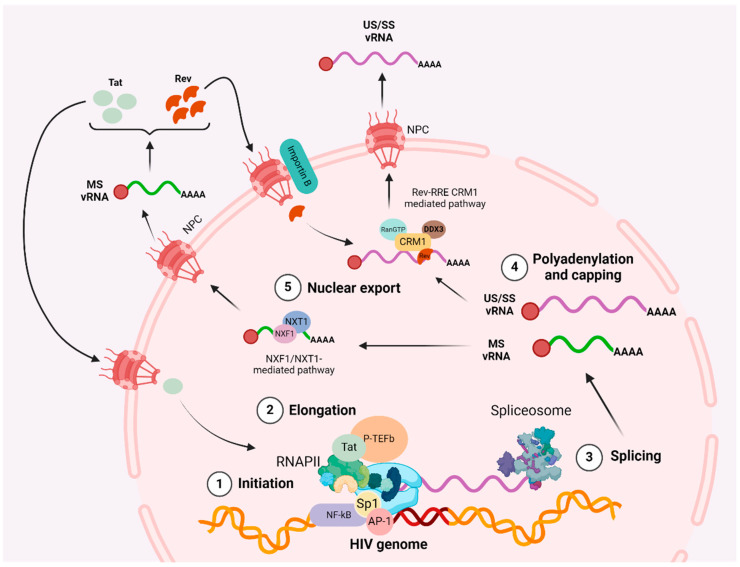
HIV post-transcriptional steps. HIV transcription initiation depends on the balance of certain host transcription factors (activators or repressors). Once initiated, Tat and P-TEFb are responsible for recruiting and activating RNA polymerase II (RNAPII), which will start HIV elongation. During HIV elongation, the spliceosome will be recruited around the RNA generated and start the splicing process, generating three major products: unspliced HIV vRNA (US vRNA) that serves as new HIV virion genome; singly spliced HIV vRNA (SS vRNA), used to produce structural and accessory viral proteins and multiply spliced HIV vRNA (MS vRNA), essential to produce Tat and Rev proteins. Then, the two vRNA will be capped and polyadenylated by the cell machinery. The last step is the HIV vRNA nuclear export; on the one hand, MS vRNA will be exported via the NFX1/NXT1 pathway through the nuclear pores (NPC). As MS vRNA is transported to the cytoplasm it will start the translation step in order to produce Rev and Tat. Lastly, Tat can return to the nucleus to enhance viral transcription, while Rev will also return to the nucleus via Importin B and bind to the RRE to export US vRNA to the cytoplasm via the canonical CRM1-mediated pathway.

## 3. Pharmacological Disruption of HIV Transcription and Nuclear Export

On the one hand, many latency-reversing agents (LRAs) have been investigated for their efficacy in inducing virus production from latently infected CD4+ T cells in vitro and in vivo [56,57,58,59,60]. Among the LRAs tested, histone deacetylase inhibitors (HDACi) induce chromatin decondensation, BET bromodomain inhibitors (BETi) release of the positive transcription elongation factor b (P-TEFb), and small-molecule antagonists of cIAP1 (SMAC mimetics) activate the non-canonical NF-κB signaling pathway, which leads in all three cases to subsequent relief of transcriptional repression in the 5′LTR and activation of HIV expression [61,62,63,64,65,66,67]. However, while LRAs such as HDACi have been shown to cause modest in vivo increases in cell-associated HIV transcripts (including initiated, 5′elongated, unspliced, and/or polyadenylated HIV transcripts) and transient increases in plasma viremia [60], they failed to increase multiply spliced HIV RNA (msRNA) in vitro or in vivo, which is a marker of productive infection [68,69,70]. Importantly, these LRAs also failed to induce a sustained reduction in the frequency of latently infected cells in vivo. These results may be explained because HIV expression is also further restricted by inefficient splicing [71] and possibly defects in the nuclear export of HIV RNA [72]. One interesting target for LRAs is Tat, essential for HIV vRNA elongation. Recent work showed a potent viral reactivation with the use of nanoparticles containing a truncated variant of the Tat protein, known as T66 [73,74]. Nanoparticles containing Tat RNA used in previous works showed that using Tat protein was not that effective, as Tat is known to become trapped in endosomes [75]. Another potential family of compounds that may act as LRAs are small molecule mimetics of the ubiquitin ligase BIRC2 (cIAP1) [76], a repressor of the noncanonical NF-kB pathway, which increased HIV-1 expression without mediating systemic T cell activation. For example, ciapavir induced activation of HIV-1 reservoirs in vivo in a humanized mouse model [67], and AZD5582 increased HIV- and SIV-RNA expression in the blood and tissues of ART-suppressed humanized mice and rhesus macaques [66].

On the other hand, novel compounds, known as latency-promoting agents (LPAs), have recently been developed to permanently shut off the transcription of HIV [7,77,78]. LPAs include different compounds that, by targeting different viral or cellular proteins, impair HIV-1 reactivation. Didehydro-Cortistatin A (dCA) [8,11,79], a Tat inhibitor, suppresses SIV replication and reactivation [80] and prevents HIV-1 reactivation from latency [81]. Unfortunately, Tat inhibitors are not clinically available, and a new generation of chemical derivatives, which bind and inhibit an active and specific Tat conformer, is needed [11]. Tyrosine kinase inhibitors [82], such as dasatinib, which, by preserving SAMHD1 activity, protected humanized mice from acute HIV-1 infection [83] and disturbed the HIV-1 reservoir reactivation and reseeding in ART-suppressed individuals [82]. Manidipine hydrochloride, a suppressor of gene expression noise, substantially reduced latent HIV-1 reactivation in vitro [12]; therefore, modulators of gene expression noise could be used in strategies to limit spontaneous reactivation of latent HIV-1. This is also the case of inhibitors of thioredoxin reductase, a protein involved in maintaining redox balance in host cells, which were discovered by screening for gene expression noise, and they promote HIV-1 latency in vitro [7]. Senexin A is an inhibitor of the cyclin-dependent protein kinases CDK8/CDK19, which are required for expression from 5′HIV-1 LTR, and suppresses proviral reactivation in vitro [84].

An interesting HIV transcription stage to be tackled by LPAs could be the polyadenylation step, although its principal regulation by the cellular machinery suggests that its approach could be cytotoxic. Nonetheless, a few studies have proposed interventions to disrupt vRNA polyadenylation. Particularly, a study showed that a mutation in the splice donor site promoted the activation of a cryptic polyadenylation site (CpA), causing decreased HIV transcriptional activity [85].

Additional targeting processes for both HIV latency reactivation and silencing is the splicing step. In this regard, few compounds have been investigated as potential LRAs or LPAs. A study showed that digoxin, a drug widely used in congestive heart failure treatment [86], suppresses HIV-1 replication by altering viral RNA processing [87], while another study suggested that Filgotinib, a Janus kinase (JAK) inhibitor, suppresses HIV-1 transcription by inhibiting T cell activation and by modulating RNA splicing [88]. In-depth investigation and high-throughput screening of potential compounds impacting RNA splicing should be performed to find novel efficient LRAs or LPAs that may modulate this HIV transcription stage.

The nuclear export step constitutes the last step of the transcription that can be perturbed. Because the adequate balance between US and SS vRNA depends on the Rev–RRE interaction, many LPAs tackle this specific step through mechanisms that involve inhibiting Rev from returning to the nucleus [89,90] or disrupting the Rev–RRE interaction [91,92,93,94]. The advantage of intervening in the Rev–RRE interaction is its viral specificity, which, in principle, would limit toxicity. Therefore, the Rev–RRE interaction or the Rev–CRM1 interaction has been widely studied, and several LPAs have been described, such as Leptomycin B [91], which targets the Rev–CRM1-mediated export, or Benfluoron and its analogs, which target the Rev–RRE interaction through binding to RRE [92,95]. Furthermore, it has been shown that Ivermectin can disrupt the nuclear import of Rev by inhibiting Importin-mediated nuclear import and thereby preventing the export of SS and US vRNA by Rev [89]. Despite the fact that the above-mentioned compounds have demonstrated effective antiviral effects in vitro, their intrinsic toxicity has hindered their use in clinical trials [92,95]. Finally, obefazimod (formerly context, the small molecule ABX464) showed important inhibition of HIV replication by binding to the Cap binding complex (CBC) and preventing Rev–RRE mediated transport of US vRNA to the cytoplasm, also altering its splicing and reducing HIV initiation and elongation [96,97,98]. Still, looking for novel compounds targeting the Rev-mediated nuclear transport is an active area of research. A summary of the LRAs and LPAs evaluated in vitro, in vivo, and in clinical trials is shown in Table 1 and Table 2, and in Figure 2.

## 4. Genetic and Epigenetic Modulation to Impact HIV Transcription

On the one hand, the development of gene editing tools, including zinc finger nucleases (ZFNs) [111], transcription activator-like effector nucleases (TALENs) [112], and the CRIPSR-Cas9 system [113], has created novel strategies to tackle viral latency in the field of HIV persistence. These advancements may include (a) directly targeting the virus in its integrated or non-integrated forms, (b) modulating its transcription as a “block and lock” [114] or “shock and kill” [115] strategies by using catalytically inactive Cas9 [116,117], (c) targeting host genes involved in the viral replication cycle (such CCR5) [118], and (d) discovering novel host molecular genes involved in the viral replication cycle using CRISPR as a screening tool [119].One of the novelties brought by the CRISPR-Cas9 system is the simpleness of the model, as CRISPR-Cas9 technology uses RNA as a guide to target directly objective genes, making it easier to design than ZFNs and TALENs. CRISPR-Cas9 is being widely used in the HIV field in applications ranging from the modulation of HIV transcription [114] to the discovery of host genes involved in the viral replication cycle [119]. Currently, there are two ongoing clinical trials using CRIPSR-Cas9 in HIV. However, only one (NCT05144386) has a recorded status. In this trial, the aim is to evaluate EBT-101, which is designed to excise HIV proviral DNA using CRISPR-Cas9 and two guide RNAs (gRNAs) targeting LTRs and Gag regions and is delivered to cells as a one-time treatment via an adeno-associated virus (AAV), administrated intravenously to aviremic adults on stable antiretroviral treatment. EBT-001 (a homolog of EBT-101 to target SIV) has already been successfully used to eliminate integrated SIV DNA in non-human primates [120]. EBT-001, similar to EBT-101, consists of an adeno-associated virus serotype 9 (AAV9) encapsulated in an all-in-one CRISPR construct that simultaneously expresses the SaCa9 endonuclease and dual gRNAs that targets LTRs and the Gag regions. Therefore, both EBT-101/001 can generate three different deletions in the HIV/SIV integrated genome: 5′LTR to Gag, Gag to 3′LTR, and 5′LTR to 3′LTR. Furthermore, in a recent study, EBT-001 showed no off-target effects or abnormal pathology in non-human primates (macaques) [120]. These positive results prompted a clinical trial, which is currently ongoing (NCT05144386). Several studies have evaluated the applicability of catalytically inactive Cas9 to reactivate or silence HIV and SIV transcription [114,121,122,123,124,125,126,127,128], but all these studies are still in the pre-clinical stage.

On the other hand, the discovery of new compounds that are integrase inhibitors and contribute to HIV integration site selection has opened novel potential mechanisms to regulate HIV transcription. In this regard, LEDGINs are small-molecule integrase inhibitors that target the binding pocket of LEDGF/p75, a cellular cofactor that substantially contributes to HIV integration site selection. These compounds are potent antivirals that inhibit HIV integration and maturation, but, in addition, they lead pre-integrated HIV DNA away from transcriptionally active regions and towards a more repressive chromatin environment of the cellular genome. This modulation of the HIV integration site demonstrated the establishment of more latent and more refractory reactivation proviruses in vitro [129], supporting the use of these compounds as potential indirect LPAs.

Furthermore, microRNAs (miRNAs) are small non-coding RNAs that bind mRNAs based on sequence complementarity to regulate protein expression as a mechanism of post-transcriptional epigenetic regulation [130,131,132]. It has been extensively demonstrated that cellular RNA interference machinery, such as miRNAs, play key roles in controlling viral infections [133,134]. Several cellular miRNAs have been shown to play a direct or indirect role in modulating HIV-1 replication (reviewed in [135,136]). Specifically, focusing on HIV transcriptional and post-transcriptional regulation, there is not much research published yet. A study showed that miR-29, which directly targets the 3′UTR of viral mRNAs, can bind HIV-1 mRNA and increase its interaction with proteins involved in post-transcriptional processes, such as mRNA degradation, inhibiting translation of viral proteins and viral replication [137]. Other studies found that miR-27b, miR-29b, miR-150, miR-198, and miR-223 inhibit the expression of Cyclin T1, an important component of the eukaryotic RNA polymerase II elongation complex, and reduce its protein levels in different cell types, which decreases HIV-1 transcription [138,139]. Another study observed that the miRNA cluster miR-17/92 targets p300-CREB binding protein associated factor (PCAF), which is important for Tat acetylation and HIV-1 LTR-driven transcriptional up-regulation, thus decreasing the efficiency of HIV-1 transcription [140]. Hence, the use of specific miRNA mimics or inhibitors might be an attractive novel strategy to impact HIV-1 transcriptional and post-transcriptional regulation and impact latency.

A schematic representation of pharmacological, genetic, and epigenetic modulation of HIV transcription is shown in Figure 2.

**Figure 2 microorganisms-12-00752-f002:**
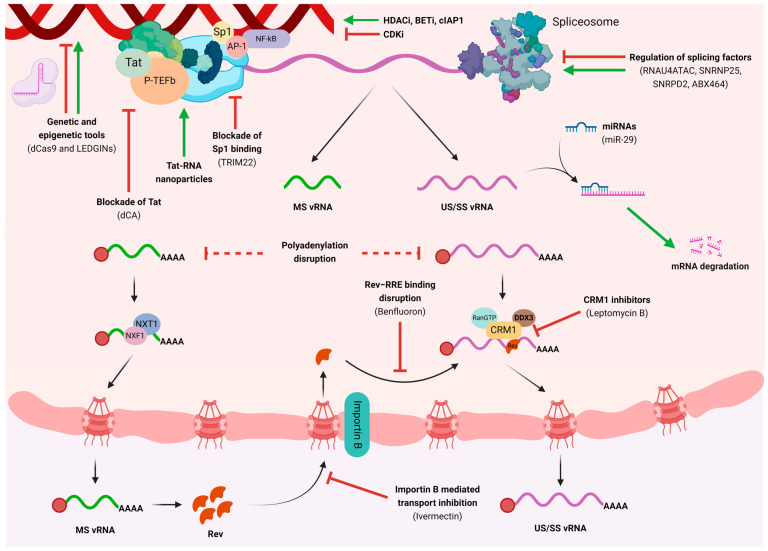
Graphical representation of host factors, compounds, and tools that regulate HIV transcription and their main targets.

## 5. What Can We Learn from HIV-2?

The two main subtypes of HIV, HIV-1 and HIV-2, are differentiated by their replicative and pathogenic capacity, virus evolution, and target of infection. HIV-1 is the most prevalent type of HIV spread worldwide, whereas HIV-2 is mostly found in West Africa. This different distribution might be explained by a lower capacity for transmission [141]. Furthermore, although both variants share similarities in transmission routes and can cause acquired immunodeficiency syndrome with comparable clinical manifestations, HIV-2 infection is generally milder and less likely to progress to AIDS. Thus, as HIV-2 is naturally less pathogenic, it might be used as a model to provide insight into alternative HIV cure strategies.

Most HIV studies are focused on the HIV-1 variant; however, HIV-2 should not be neglected as an estimated 2 million people worldwide are living with this infection, and there might be differences in the mechanisms that regulate HIV-2 persistence and latency. Recently, Koofhethile et al. observed that the HIV-2 proviral landscape is dominated by defective proviruses, similar to what happens in HIV-1 infection [142]. Nonetheless, although HIV-2 LTR is similar to HIV-1 LTR, it contains a duplicated TAR RNA stem-loop structure [143] and is less responsive to CD4+ T cell activation signals [144]. These differences in transcriptional control of HIV-2 might be due to differential regulation affecting basal transcription levels and the response to environmental transcription regulators. In that context, as previously mentioned in Section 2, TASOR is a repressive transcriptional factor of HIV-1, but interestingly, Vpx from HIV-2 induces its degradation [55], suggesting that this might be a differential mechanism of HIV-1 and HIV-2 transcription regulation. Furthermore, Pedro et al. [48] identified several transcription factors that preferentially bound HIV-2 LTRs. Interestingly, they identified PLAGL1, which was not previously described as a regulator of HIV transcription, as an HIV-2-specific transcriptional activator [48]. This transcription factor is widely expressed in immune cells, recognizes GC-rich DNA regions, has both transactivating and repressing activities [145,146,147], is involved in cell cycle regulation and oncogenesis [147,148], and interacts with other transcriptional activators of HIV, such as SP1, AP-1, and PCAF/CBP/P300 [148]. This study suggested that HIV-2 LTR is differentially regulated compared to HIV-1 and that there is a need for a more detailed transcriptional analysis of the mechanisms that control HIV-2 proviral expression [48]. In this regard, Lu et al. [149] developed a panel of novel PCR assays to investigate the mechanisms of latent infection with HIV-2, showing that HIV-2 transcription is regulated by blocks to elongation and completion in peripheral CD4+ T cells [150]. Hence, further characterization of the mechanisms regulating HIV-2 transcription in vivo is essential to find a worldwide applicable HIV cure strategy.

## 6. Conclusions

There is a lack of knowledge on how RNA metabolism can be used as a strategy to cure HIV. In this review, we summarize the cellular and viral factors that regulate HIV-1 transcription, the potential pharmacological targets and genetic and epigenetic editing techniques that have been or might be evaluated to disrupt HIV-1 latency, the role of miRNA in post-transcriptional regulation of HIV-1, and the differences between the mechanisms regulating HIV-1 and HIV-2 expression. The in-depth investigation of these pathways, drugs, and techniques is essential to increase our current knowledge of HIV latency and persistence and will potentially lead to novel therapeutic targets advancing to a global HIV cure strategy.

## Figures and Tables

**Table 1 microorganisms-12-00752-t001:** List of LRAs evaluated in pre-clinical or approved status.

Compound	Family of Compounds	Type	Target	Clinical Status	References
Vorinostat	Histone deacetylase inhibitors (HDACi)	LRA	Histone deacetylase	FDA approved	[56,57,63,69,99]
Romidepsin	Histone deacetylase inhibitors (HDACi)	LRA	Histone deacetylase	FDA approved	[58,60,65,70,99]
Panobinostat	Histone deacetylase inhibitors (HDACi)	LRA	Histone deacetylase	FDA approved	[10,59,65,69,99]
Belinostat	Histone deacetylase inhibitors (HDACi)	LRA	Histone deacetylase	FDA approved	[99]
Apabetalone (RVX-208)	Bromodomain inhibitors (BETi)	LRA	BD2/BRD4	Pre-clinical	[100,101,102]
CPI-203	Bromodomain inhibitors (BETi)	LRA	BRD4	Pre-clinical	[103]
I-BET-151	Bromodomain inhibitors (BETi)	LRA	BD1, BD2/BRD4	Pre-clinical	[62,104]
MMQO	Bromodomain inhibitors (BETi)	LRA	BRD2-4/BRDT	Pre-clinical	[105]
OTX-015	Bromodomain inhibitors (BETi)	LRA	BRD2-4/BRDT	Pre-clinical	[106,107]
PFI-1	Bromodomain inhibitors (BETi)	LRA	BRD2/BRD4	Pre-clinical	[100]
UMB-136	Bromodomain inhibitors (BETi)	LRA	BD1/BRD4	Pre-clinical	[108]
Ciapavir	Small molecules mimetic of cIAP1	LRA	cIAP1	Pre-clinical	[67]
SBI-0637142; Debio-1143, AZD5582	Small molecules mimetic of cIAP1	LRA	cIAP1	Pre-clinical	[76]
Nanoparticles containing Tat mRNA (T66)	Tat agonists	LRA	Tat	Pre-clinical	[73,74]

**Table 2 microorganisms-12-00752-t002:** List of LPAs evaluated in pre-clinical or approved status.

Compound	Family of Compounds	Type	Target	Clinical Status	References
didehydro-Cortistatin A (dCA)	Tat inhibitors	LPA	Tat	Pre-clinical	[11]
Dasatinib	Tyrosine kinase inhibitors (TKIs)	LPA	SAMHD1	Pre-clinical	[109]
Manidipine hydrochloride	Noise suppressor of gene expression	LPA	Calcium channel blocker	Pre-clinical	[12]
TE-2, TE-10, TE-14, TE-20	Thioredoxin reductase inhibitors	LPA	Thioredoxin reductase redox pathway	Pre-clinical	[110]
Senexin A	Cyclin-dependent protein kinases inhibitors (CDKi)	LPA	CDK8/CDK19	Pre-clinical	[84]
Filgotinib	Janus kinase (JAK) inhibitor splicing modulator	LPA	JAK HIV mRNA	FDA-approved	[88]
Digoxin	Splicing modulator	LPA	HIV mRNA	FDA-approved	[87]
Leptomycin B	Nuclear export inhibitor	LPA	CRM1	Pre-clinical	[91,93]
Benfluoron	Nuclear export inhibitor	LPA	Rev-RRE	Pre-clinical	[92]
Ivermectin	Nuclear export inhibitor	LPA	Importin B	Pre-clinical	[89]

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
