# Peer review of "Targeting Viral Transcription for HIV Cure Strategies"

_microorganisms, 2024, doi:10.3390/microorganisms12040752_

Round 1

Reviewer 1 Report

Comments and Suggestions for Authors

The authors summarize the knowledge about the cellular and viral factors that regulate HIV-1 transcription, the potential pharmacological targets and genetic and epigenetic editing techniques to disrupt HIV-1 latency, and the differences between the mechanisms regulating HIV-1 and HIV-2 expression. Overall, I find this review to be informative, of high quality, and up-to-date. I do not have any major concerns regarding the content. 

Comments on the Quality of English Language

I suggest some language editing, for example "HIV mRNA follows the same polyadenylation process that other cellular mRNAs." should be changed to HIV mRNA follows the same polyadenylation process as other cellular mRNAs.". Also, chromatin appears to be misspelt as "chromatine" in the text.

Author Response

Thank you very much for taking the time to review this manuscript. Please find the detailed responses in the attachment and the corresponding revisions/corrections in track changes in the re-submitted files.

Reviewer 2 Report

Comments and Suggestions for Authors

The review article by Inquierdo-Pujol et al comprehensively describes the current state of cure strategies targeting HIV transcription and replication. The manuscript is well-written, covers relevant citations and the figures are informative. 

However, if the authors want to include the role of miRNAs in HIV transcription, the article will benefit from further expanding on this section. The section 5 is underdeveloped and barely touches on the subject with mentions of a few review article. Otherwise, if deemed beyond the scope of this article, then the section 5 should be removed altogether.

Similarly, section 6 is also underdeveloped with minimal background and inference on the impact of HIV-2 based research on HIV-1 therapy, provided they have different regulation of viral transcription. Minimal relevance to the overall topic. 

Table 1 can be made more concise by combining the drug compounds that target the same molecule into a single row. LRAs and LPAs should be distinguished either by color or separating into two tables. 

Comments on the Quality of English Language

Minor editing for grammatical correctness.

Author Response

(The authors gave the same response as above.)
